# Influence of Pre-Harvest Bagging on the Incidence of *Aulacaspis tubercularis* Newstead (Hemiptera: Diaspididae) and Fruit Quality in Mango

**DOI:** 10.3390/insects12060500

**Published:** 2021-05-27

**Authors:** Modesto del Pino, Claudia Bienvenido, María Eva Wong, María del Carmen Rodríguez, Juan Ramón Boyero, José Miguel Vela

**Affiliations:** Instituto de Investigación y Formación Agraria, Pesquera, Alimentaria y de la Producción Ecológica (IFAPA), Centro Málaga, Cortijo de la Cruz s/n, 29140 Churriana, Malaga, Spain; claudia.bienvenido@juntadeandalucia.es (C.B.); mariae.wong@juntadeandalucia.es (M.E.W.); mcarmen.rodriguez.lopez@juntadeandalucia.es (M.d.C.R.); juanr.boyero@juntadeandalucia.es (J.R.B.); josem.vela@juntadeandalucia.es (J.M.V.)

**Keywords:** IPM, cultural control, white mango scale, fruit quality, *Mangifera indica*

## Abstract

**Simple Summary:**

The white mango scale *Aulacaspis tubercularis* is an invasive pest that causes important damage to mango crops in Southern Spain. The use of integrated management strategies (cultural, biological, and chemical control) is recommended for its effective and sustainable management. Among cultural control measures, fruit bagging technology is being widely used in some mango production regions prior to harvest to protect fruit from diseases, pests, and mechanical damage. However, despite the proven efficacy of bagging, its usefulness has still not been proved to control *A. tubercularis* infestations on mango fruits. In this study, we evaluated not only the mango bagging efficacy on *A. tubercularis* control but also its effects on the development and quality of bagged fruits. We tested two commercial types of bags (a yellow satin paper bag and a white muslin cloth bag) commonly used to cover several other fruits. Both bags were effective in reducing the pest incidence and damage caused by *A. tubercularis* when mango fruits were bagged before the scales migrated to them. Our findings indicate that the use of bags for the pre-harvest protection of mango fruits is feasible, and it may improve their development (weight and size) and quality (color and soluble solids).

**Abstract:**

*Aulacaspis tubercularis* Newstead (Hemiptera: Diaspididae) is the main pest of mango, *Mangifera indica* L., in Spain, causing significant economic losses by aesthetic damage that reduce the commercial value of fruit. Bagging fruit with two commercial bags (a yellow satin paper and a white muslin cloth bag) was evaluated for control of *A. tubercularis* in two organic mango orchards during the 2020 cropping season in pursuit of the development of a mango IPM program to produce pest-free and residue-free fruits. Results from fruit damage evaluations at harvest showed that bagging significantly reduced pest incidence and fruit damage compared with non-bagged plots. Of the two bags evaluated, white muslin cloth bag provided higher levels of fruit protection from *A. tubercularis* damage, reducing the non-commercial fruit percentage by up to 93.42%. Fruit quality assessment indicated that weight and size of bagged fruit were significantly higher than the non-bagged. Paper-bagged mangoes showed higher whiteness and yellowness compared to the other treatments. Soluble solids content (ºBrix) was higher in paper-bagged fruit than all other treatment plots. The results from this study indicate that pre-harvest fruit bagging is effective at controlling *A. tubercularis* and should be integrated into an IPM program for Spanish mango production.

## 1. Introduction

Mango (*Mangifera indica* L.) is a tropical fruit tree native to Southeast Asia belonging to the Anacardiaceae family, which is grown commercially in tropical and subtropical regions of the world [1]. Currently, mango is one of the most consumed fresh fruits worldwide (along with bananas, oranges, grapes, and apples) with a global production that reached more than 56 million tons in 2019 [2]. A few countries (India, Indonesia, China, Mexico, Pakistan, Malawi, and Brazil) account for over 75% of world production, India being the main mango-producing country, with about 25 million tons [2]. Mango cultivation area has been significantly extended in the last decades, even in regions far away from the equator such as several countries of the Mediterranean basin, including Egypt, Israel, Spain, and Italy [1,3,4]. This has been due to the high demand for this tropical fruit among consumers because of its attractive fragrance, beautiful color, taste, and nutritional properties, being an abundant source of vitamins and minerals [5]. At present, southern Spain is the only region in mainland Europe with a significant commercial mango production with a current extension of more than 5300 ha and a total production of about 32,200 t in 2019 [6].

Until a few years ago, Spanish mango cultivation was characterized by a low presence of problems caused by pests and diseases [7,8]. However, the increase in the international trade, mainly of live plants, has facilitated the introduction and establishment of several exotic pest species in the last ten years [7,9] such as the white mango scale, *Aulacaspis tubercularis* Newstead (Hemiptera: Diaspididae). This armored scale is currently the main pest in the Spanish mango orchards and one of the most important pests of this crop worldwide in tropical and even subtropical regions [8]. *A. tubercularis* causes conspicuous pink blemishes on the epidermis of the ripe mango fruits [10,11], which affect their commercial value [12,13]. Some estimates suggest that *A. tubercularis* can cause important economic losses in the Spanish organic mango orchards, which may exceed 40% in late-ripening cultivars [14]. Control of *A. tubercularis* is mainly carried out through the mango tree post-harvest pruning and the repeated application of a limited number of authorized insecticides during the crop cycle, which may increase the risk of resistance development, cause a negative impact on beneficial insects and the environment, and generate pesticide residues that make the marketing of the mango fruits difficult [8,14]. Thus, development of Integrated Pest Management (IPM) strategies, based on the conservation of natural enemies and the use of low-impact insecticides, can mitigate the economic losses caused by *A. tubercularis* in the Spanish mango orchards [8,14,15].

Pre-harvest fruit bagging is a traditional Asian physical protection method commonly applied to many fruits [16,17]. This technology has been employed to minimize pest damage to several fruits such as mangoes, guavas, lychees, pomegranates, citrus, and pitayas [17]. In recent years, fruit bagging technique is being used in the major mango growing regions prior to harvest to protect fruit from diseases (anthracnose and stem rot), pests (fruit flies and other insects), and scratches, to improve skin color, and to reduce sunburn, fruit cracking, agrochemical residues, and bird damage [8,16,17,18,19]. Different studies indicate that fruit bagging also improves the internal quality of mango fruit [17,20,21,22,23] due to the micro-environment created by the bag around the fruit [18]. In addition, this technique increases the marketable yield, the size and weight of bagged fruit being higher than those that are unbagged [24]. Despite these advantages, further research is still needed to determine which type of bag is most appropriate for the different mango varieties, as well as the best time to bag the fruit [21].

Several studies have reported the bagging technique as a successful control measure against fruit flies in commercial mango orchards [16,18,25,26,27,28,29]. Studies conducted in Bangladesh revealed that infestation levels of the Oriental fruit fly, *Bactrocera dorsalis* (Hendel) (Diptera: Tephritidae), were significantly reduced up to 100% by covering the fruits with double-layer brown paper bags at least 30 days prior to harvest [18,29]. In Mexico, the bagged Manila cv. mango fruit showed 100% sanitation from the fruit fly *Anastrepha obliqua* (Macquart) (Diptera: Tephritidae), 90% control of anthracnose (*Colletotrichum gloeosporioides* Penz.) in ripe fruit, and a considerable reduction of damage caused by *Capnoium* spp. [26]. Bagging has also been shown to be an effective option for the organic control of pests such as the mango seed weevil, *Sternochetus mangiferae* (Fabricius) (Coleoptera: Curculionidae), and some scale and mealybugs in South Africa [30]. In this sense, experiments carried out by Chonhenchob et al. [24] in Thailand reported that bagged mangoes showed less damage caused by anthracnose, insects, animals, cuts, abrasion, and skin browning than unbagged mangoes. However, according to the literature, no specific studies have been carried out to assess the efficacy of fruit bagging on reducing the damage caused by *A. tubercularis* or other species of Coccoidea. In this sense, this physical protection method may be an appropriate non-chemical strategy to incorporate into existing IPM programs for mango in Southern Spain, provided that bags are easy to set up and economically feasible, and fruit protection from scales is similar to that of chemical control.

Thus, the main objective of this study was to investigate the field efficacy of pre-harvest mango bagging to avoid damage by *A. tubercularis* and thus reduce economic losses and to evaluate physiological effects of bagging on fruit such as size and weight, skin color, internal flesh color, and total soluble solids content. Of course, mango bagging could also contribute to reducing other pests, either present as some species of coccids or even not yet present in the Southern Spain groves, which probably will colonize this area.

## 2. Materials and Methods

### 2.1. Study Site

Studies were conducted in two organic mango orchards located in Algarrobo (Malaga, Spain) during the fruiting season of 2020. The first orchard (36°45′7.0″ N, 4°3′26.5″ W; 0.40 ha) contained fifteen years old Osteen cv. mango trees, which were planted at a spacing of 4 m between rows and 3 m between trees. The second orchard (36°45′37.4″ N, 4°3′38.0″ W; 0.94 ha) contained ten years old Sensation cv. mango trees with a plantation framework of 4 m × 3 m. Both mango orchards were drip-irrigated, and no pesticides were sprayed during the sampling period, except some fungicides authorized in organic production (sulfur) to control the powdery mildew (*Oidium mangiferae* Berthet). The presence of *A. tubercularis* had been recorded in both orchards for several years before the study. Trees were maintained following the cultural practices routinely employed in this mango cultivation area.

### 2.2. Fruit Bags

Two commercial fruit bags were evaluated during the study. First one was a rectangular slightly yellowish satin paper bag (Sanidad Agrícola Econex S.L., Murcia, Spain) measuring 35.0 cm by 23.5 cm and similar to those used in the grapes industry in Spain. This bag is made of virgin cellulose and satin on the outside, opened at both ends, and has a thickness of approximately 5 µm. In our case, one end of the bag was stapled to prevent insect access. The second one was a white muslin cloth bag (Junejour New Life Store, China) measuring 30.0 cm in length by 24.5 cm in width, such as those used to cover several fruits in China.

### 2.3. Treatments and Experimental Design

A completely randomized design with three treatments and twelve single tree replications was performed. In each orchard, twelve mango fruits per tree, without presence of *A. tubercularis,* were selected and randomly assigned to the following treatments: T_1_ = yellow paper bag, T_2_ = white cloth bag, and T_3_ = non-bagged (control). Four fruits per tree for each treatment were used. In both orchards, uniformly grown fruits were bagged when they were 7–10 cm long or approximately 45 days (28 July) after fruit set. The bags were distributed uniformly around each canopy and at all heights. Bags were secured around the fruit peduncle with a thin wire or drawstring embedded along one edge of the bag. Mature fruit were harvested 71 and 91 days (7 and 27 October) after bagging for the first and second orchards, respectively. All mango fruits from each treatment and orchard were harvested. Bags were removed immediately after harvest, and fruits were labeled. The harvested fruits of each treatment were kept separately and brought to the laboratory the same day for further observations.

### 2.4. Fruit Damage Assessment

Fruit damage by *A. tubercularis* was evaluated at harvest. All selected fruit from the treatments remaining on trees at harvest were assessed. Fruits were classified as free of damage (clean) or damaged by pests. Fruit was considered damaged if it exhibited signs of individual pest infestation or damage. The surface of fruit was inspected for the presence of females and male colonies of *A. tubercularis*, as well as external lesions and conspicuous pink blemishes on the epidermis. In our case, fruit that exceeded the threshold of four pink blemishes per piece was considered as non-commercial fruit for export and fresh consumption, following the quality criteria established by the local marketing companies [1,14]. The presence of other Coccoidea and even other internal or external physical damage were also recorded.

Subsequently, the percentages of damaged and non-commercial fruits were calculated from each treatment plot. The number of fruits damaged by *A. tubercularis* was converted to proportion damaged by dividing the number damaged by the total number of fruits evaluated from each treatment plot.

### 2.5. Fruit Quality Assessment

Fruit development (weight, size, and sphericity) and quality (color and soluble solids) were assessed according to Chonhenchob et al. [24]. Size and sphericity were measured at the widest position using a Vernier caliper and expressed by the following equations:Size = (*abc*)^1/3^,(1)
Sphericity = *b*/*d*(2)
where *a* = width, *b* = length, *c* = depth, and *d* = circumference.

Peel color was determined with a CR-300 Chromometer (Minolta Inc., Osaka, Japan) using the CIE system. Data were expressed as *L** *a** *b** values, where: *L** indicates the lightness (0 = black and 100 = white); *a** represents the green and red color (+*a** = green and *−a**** = red); and *b** indicates the yellow and blue (+*b** = yellow and −*b** = blue). The hue angle (°) and chroma were estimated using the following equations:Hue angle (°) = tan^−1^ (*b**/*a**),(3)
Chroma = (*a**^2^ + *b**^2^)^1/2^(4)

Internal flesh color development stages were estimated according to the procedure described in the Mango Maturity and Ripeness Guide provided by the National Mango Board in the USA (NMB 2010), which describes five stages (1 to 5) defined by the appearance and progression of yellow to orange color, from 0 to 100% of the flesh in 20% increments starting near the pit and progressing outward. Soluble solids or sugar content was measured using a HI-96801 Refractometer (Hanna Instruments Ltd., Leighton Buzzard, UK), expressed as ºBrix.

### 2.6. Economic Analysis

A full economic analysis was conducted for the first and second organic mango orchards, comparing the costs per hectare derived from an estimated *A. tubercularis* chemical control in mango with the costs derived from the pre-harvest fruit bagging technique. For this purpose, the costs associated with the purchase of the insecticide product (price, application rate, and frequency of treatments) and its application (labor and time spent) were considered. The price of the insecticide product was obtained from local distributors and the application rate (doses/ha) was that recommended by the product label. Finally, the frequency of treatments was that used by mango growers in the area.

Adjusted net returns were calculated based on the selling price received by growers in the area during the fruiting season of 2020 for both mango cultivars (Osteen and Sensation). Costs were estimated for all treatments, including bags and labor costs. According to Hossain et al. [29], adjusted net return of treatment was calculated as: Net return of treatment–Net return of untreated control.

### 2.7. Data Analysis

All data were analyzed by analysis of variance (ANOVA) test applying the GLM procedure, and the average values were compared by Tukey’s honestly significant difference (HSD) test (*p* ≤ 0.05) by means of the statistical software IBM^®^ SPSS^®^ Statistics v26.0 for Windows (SPSS Inc., Chicago, IL, USA). All proportion data were subjected to arcsine square root transformation prior to analysis to meet the assumptions of normality and homogeneity of variances. Abbott’s formula [31] was employed to verify the efficacy of the fruit bagging on the *A. tubercularis* incidence with respect to the control.

## 3. Results

### 3.1. Fruit Damage

Significant differences in the number of females and male colonies of *A. tubercularis* and the number of pink blemishes on the epidermis were found among treatments at the harvest time (Figure 1). Both fruit bags provided significantly increased fruit protection from *A. tubercularis* than the untreated control in the first (females: *F* = 4.992; *df* = 2; *p* ≤ 0.001, male colonies: *F* = 5.278; *df* = 2; *p* ≤ 0.001) and second (females: *F* = 9.426; *df* = 2; *p* ≤ 0.001, male colonies: *F* = 8.285; *df* = 2; *p* ≤ 0.001) mango orchards. In addition, both treatments showed a significantly lower number of pink blemishes on the epidermis than the untreated control in the first (*F* = 8.400; *df* = 2; *p* ≤ 0.001) and second (*F* = 10.059; *df* = 2; *p* ≤ 0.001) mango orchards. In both orchards, there were no significant differences among two commercial fruit bags relative to the presence of *A. tubercularis* females, as well as conspicuous pink blemishes on the epidermis.

Significant differences in the percentages of damaged fruits and non-commercial fruits were found among treatments (Table 1). Both fruit bags significantly reduced the percentage of damaged fruits by *A. tubercularis* with respect to the untreated control in the first (*F* = 12.138; *df* = 2; *p* ≤ 0.001) and second (*F* = 8.720; *df* = 2; *p* ≤ 0.001) mango orchards. White mango scale pressure was similar among both orchards with infestation levels of 65.28% and 62.5% in control treatments for the first and second orchard, respectively. The bagging with white cloth bags provided the maximum damaged fruit reduction in the first (68.09%) and second orchards (60.00%). Consequently, both treatments showed significantly lower percentage of non-commercial fruits that the untreated control in the first (*F* = 9.851; *df* = 2; *p* ≤ 0.001) and second (*F* = 7.304; *df* = 2; *p* ≤ 0.001) mango orchards. In both mango groves, there were no significant differences among the two commercial fruit bags from the percentage of non-commercial fruits, but the white cloth bag provided the maximum reduction by 93.49% and 92.12% in the first and second orchards, respectively.

### 3.2. Fruit Quality

Table 2 shows the changes in weight, size, and sphericity of mangoes with different bagging materials for the first and second orchards. Mango development was not significantly different among treatments in the first orchard (weight: *F* = 1.077; *df* = 2; *p* = 0.345; size: *F* = 0.795; *df* = 2; *p* = 0.455; sphericity: *F* = 0.190; *df* = 2; *p* = 0.827). However, the results from the second orchard indicate that mangoes bagged with both fruit bags were significantly higher in weight (*F* = 7.818; *df* = 2; *p* ≤ 0.001) and size (*F* = 6.792; *df* = 2; *p* ≤ 0.001) than the untreated control. In this case, the weight and size of mango fruit bagged with yellow satin paper bag were similar to the white cloth bag treatment. Additionally, fruit bagging had no significant effect on sphericity among treatments (*F* = 0.295; *df* = 2; *p* = 0.745).

Changes in *L******** *a******** *b******** values of mango peels for different bagging treatments in both orchards are shown in Table 3. Significant differences in the *L******** and *b******** values were found among treatments at the harvest time. Both fruit bags provided significantly higher lightness or whiteness (*L******** value) than the untreated control in the first (*F* = 13.023; *df* = 2; *p* ≤ 0.001) and second (*F* = 3.950; *df* = 2; *p* ≤ 0.001) mango orchards. In addition, all treatments showed significantly higher yellowness (*b*^*^ value) than the untreated control in the first (*F* = 8.877; *df* = 2; *p* ≤ 0.001) and second (*F* = 5.385; *df* = 2; *p* ≤ 0.001) mango orchards. As compared to the other treatments, paper-bagged mangoes had the highest *L******** and *b******** values in both orchards, indicating higher whiteness and yellowness and expressing more ‘pale’ yellow as compared to the other treatments. Finally, there were no significant differences among treatments with respect to the greenness (*a^*^* value) in the first (*F* = 2.268; *df* = 2; *p* = 0.109) and second (*F* = 1.645; *df* = 2; *p* = 0.198) orchards. 

Fruit quality as measured by soluble solids content (ºBrix) was not significantly different among treatments in the first orchard (*F* = 0.701; *df* = 2; *p* = 0.499) (Figure 2). However, in the second orchard, the soluble solids content of mango fruit bagged with yellow satin paper bag was significantly higher than the white cloth bag treatment but equal to the untreated control (*F* = 3.794; *df* = 2; *p* ≤ 0.001).

### 3.3. Economic Analysis

Table 4 shows the economic analysis per hectare between the cost for an estimated chemical control and the cost for the pre-harvest bagging technique against *A. tubercularis* in both mango orchards. According to our results, the cost of treatment was lowest for the chemical control in both orchards (525 EUR/ha) and highest for the white muslin cloth bag treatment with 3766.28 EUR/ha and 6000.05 EUR/ha for the first and second orchards, respectively. However, both bags provided an increase in production and fruit quality which resulted in a higher gross return for the farmer. The value of average production per hectare, considering the different quality categories and their market prices, was highest for the white muslin cloth bag treatment with 43,048.20 EUR/ha and 58,679.25 EUR/ha for the first and second orchards, respectively. Based on the above, both bags provided the highest net return in the second orchard which resulted in an adjusted net return of 6382.68 EUR/ha and 5044.61 EUR/ha for the yellow satin paper bag and white muslin cloth bag, respectively.

## 4. Discussion

*Aulacaspis tubercularis* is a serious concern for the mango industry in Southern Spain and in the world [8,14]. Other scale insects such as *Icerya seychellarum* (Westwood) (Hemiptera: Monophlebidae), *Pulvinaria psidii* Maskell, and *Ceroplastes floridensis* Comstock (Hemiptera: Coccidae) are increasing their populations and becoming important pests in the Southern Spanish mango groves [9]. The Mediterranean fruit fly, *Ceratitis capitata* (Wiedemann) (Diptera: Tephritidae), is generally not very attracted to mango fruits; however, its incidence in Southern Spain depends on the cultivar and the ripening stage at harvest [33] and may damage up to 30% of the fruits, mainly in late cultivars [32].

The most extended mango cultivar in Southern Spain is Osteen, which occupies 75% of the cultivated mango area and is collected at medium season. The other studied cultivar was Sensation, which is seldom planted and is collected in late season. Therefore, Sensation cv. fruits are more vulnerable to *A. tubercularis* and other pests [34]. Due to the more probable presence of pests, bagging technology is likely more useful when treating a late cultivar. As shown, bagging significantly reduced fruit damage by presence of *A. tubercularis* or pink blemishes produced by the pest and other scales in the two studied cultivars (Figure 1). In addition, the percentages of damaged fruits were also reduced by at least 60% and the reduction of the non-commercial fruits by more than 92% (Table 1). Although there is no published information concerning the control of *A. tubercularis* by using fruit bagging, the benefits of this technique have been demonstrated for controlling other mango pests, such as mealybugs [35], weevils (*S. mangiferae*) [30], and fruit flies in America (*Anastrepha* spp.) [36] and Asia (*Bactrocera* spp.) [18,29,37]. Our results are in accordance with those obtained by Islam et al. [35], who found that pre-harvest bagging reduced the incidence of mealybugs in mango. However, Watanawan et al. [21] found a slight increase in mealybugs on the bagged fruits. This negative effect could be resulting from a late bagging time or a deficient set-up of the bag. On the contrary, Joyce et al. [38] found a reduced blemish on Sensation cv. fruit bagged with paper bags.

In the first of the studied orchards (Osteen cv.), we did not find differences in weight or volume between controls and bagged fruits, but in the second orchard (Sensation cv.), we found that bagged mangoes were 9.6% (cloth bags) and 11.1% (paper bags) heavier than controls but only by about 3% greater in volume, given that sphericity was not different among the treatments (Table 2). These results are consistent with those previously reported by several authors who indicated that bagging increases fruit weight and total yield [17,24,29,35,39,40].

Peel color was significantly different in bagged fruits than in controls. In this sense, *L** and *b** values were higher in bagged fruits, which showed better luminosity and higher yellowish color, in the two studied cultivars. However, hue angle (color tone) was not different between treatments, and chroma value (intensity) was only slightly lower in cloth bags and higher in paper bags with respect to control in Sensation cv. Moreover, internal flesh color was not different in Sensation cv., but bagged fruits in Osteen cv. showed a higher ripening color (Table 3). Skin color is an important factor in the selling price of the fruit. The European consumers prefer mango fruits with a high percentage of red coloration in the epidermis [32]. Watanawan et al. [21] found *L** and *a** values higher in bagged mangoes cv. Nam Dok Mai #4 than non-bagged, *b** value not being different. Hoffman et al. [16] also described an increase in the skin area with yellow color and a decrease in reddish area in Keitt cv., whereas Anwar Rataul cv. fruits bagged with brown paper bags showed a more yellowish color than control or butter paper bags [40]. Finally, Wu et al. [41] found that lightness and chroma were remarkably higher, while hue angle was significantly lower than controls, and that a single white bag could be a promising practice for improving coloration.

Relative to total soluble solids (TSS) contents, only paper bags in Sensation cv. showed slightly increased ºBrix values than in controls or in cloth bags (Figure 2). Hoffmann et al. [16] and Watanawan et al. [21] did not find changes in TSS, while Hossain et al. [29] found a lesser TSS in Amrapali cv. with white paper bags. Islam et al. [35] found a higher TSS in bagged Mollika cv. either with brown paper, white paper, or muslin cloth bags, and Wu et al. [41] described higher TSS in Zill cv. bagged with single paper bags.

As seen, the literature reports different results referring to the variables treated in the present research because they may depend on the cultivar, bagging time, and the climate associated with geographic area. Generally, a good control of pests and diseases and an improvement of the appearance (including peel color) and other physical or chemical features of bagging fruits have been pointed out [18,26,39,42,43].

Our objective was to demonstrate that pre-harvest fruit bagging may be a viable and useful cultural control measure for *A. tubercularis*, without a decrease in fruits quality. Moreover, this technique may be used to control other well-known fruit pests such as the fruit fly *C. capitata* or pests with an increasing presence in Southern Spain mango groves such as *I. seychellarum*, *P. psidii*, and *C. floridensis*, and, in general, other future pests in groups such as scales (Coccoidea), fruit flies (Diptera), or fruit borers in Coleoptera and Lepidoptera. Sometimes, when fruit bags were removed in the field, specimens of the common earwig *Forficula auricularia* L. (Demaptera: Forticulidae) appeared. They use the bags for shelter [44], and their role as omnivorous insects is not exactly assessed as they may cause injury to fruits but also act as active predators for a variety of pests, including many Homoptera [45,46,47]. The status of earwigs as pests or predators is subject to dispute [48]. In our experiments, we did not observe any signal of scarring due to earwigs feeding on the fruit epidermis.

According to Ali et al. [23], fruit bagging is a relatively simple, secure, and environmentally friendly practice that reduces pest and disease damage and improves the appearance and physicochemical properties of the fruit. However, this cultural control measure can be an expensive pest management option on commercial orchards due to the time and labor required to set the bags on the fruit [17,29,44]. Although this technology may be cost-effective in developing countries [49,50], its use will depend on the mango orchard particularities, pest levels, labor costs, and selling prices received by the growers [29]. In addition, another very important factor in bagging is that it may increase the size of the mangoes compared to those non-bagged, which would allow bagging mangoes to be sold at a higher price [29]. In this sense, as our results suggest, fruit bagging is a physical technique that can be a very cost-effective and successful component of IPM for those small-scale Spanish mango producers who sell their organic fruit in selected European markets and expect to receive better prices for more quality and environmentally sustainable production [23,51].

Future research dealing with bagging time setting and removing and bagging types on different cultivars and different pests will shed more light on this interesting question which has a high impact on sustainable mango management in Spain and across the world.

## 5. Conclusions

This research showed that pre-harvest fruit bagging can reduce the incidence and damage caused by *A. tubercularis* and other mango pests in Southern Spain, in addition to improving the development and quality of the fruit. However, this pest management technique requires considerably more time and labor, being an economically unprofitable practice for large-scale mango producers in Southern Spain but very cost-effective for those small-scale organic growers who expect to sell a high quality, healthy, and sustainable fruit in selected markets.

## Figures and Tables

**Figure 1 insects-12-00500-f001:**
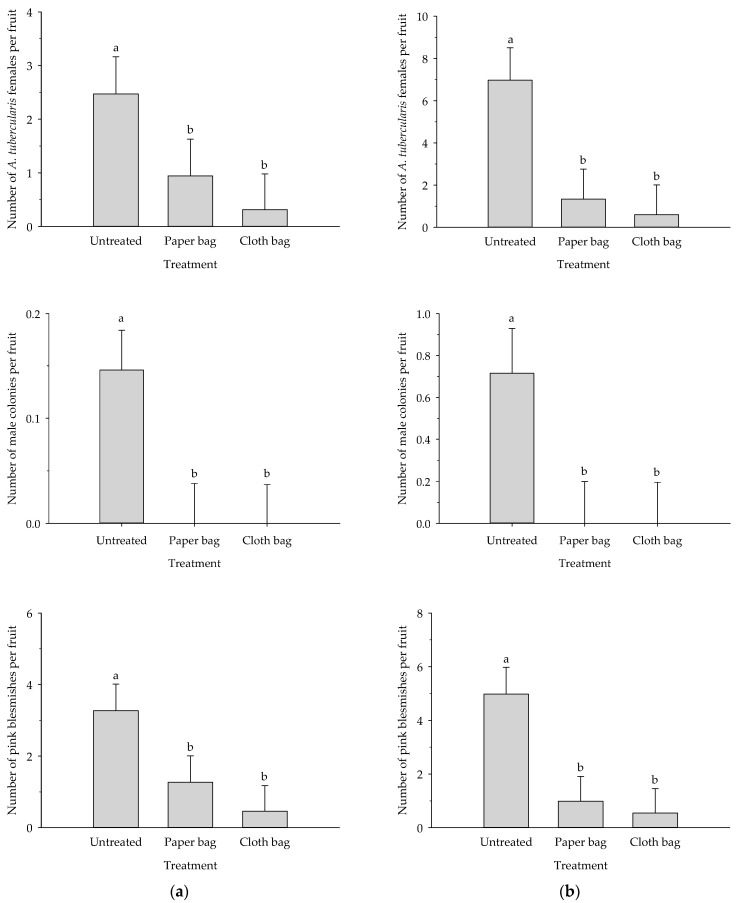
Mean (±SE) number of females and male colonies of *A. tubercularis* and pink blemishes per fruit in first (**a**) and second (**b**) mango orchards. Bars with different letters denote significant differences among treatments (Tukey’s HSD test, *p* ≤ 0.05).

**Figure 2 insects-12-00500-f002:**
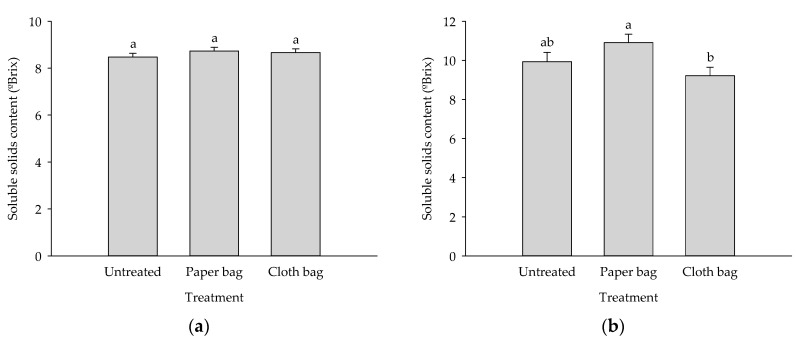
Mean (±SE) soluble solids content (ºBrix) of mango fruit in first (**a**) and second (**b**) mango orchards. Bars with different letters denote significant differences among treatments (Tukey’s HSD test, *p* ≤ 0.05).

**Table 1 insects-12-00500-t001:** Mean percentage (±SE) of fruit damaged by *A. tubercularis* and non-commercial fruits in first (a) and second (b) mango orchards.

Mango Orchard ^1^	Treatment	Damaged Fruits (%)	Infestation Reduction (%)	Non-Commercial Fruits (%)	Non-Commercial Fruits Reduction (%)
1	Untreated	65.28 ± 6.86 a	-	31.94 ± 4.95 a	-
	Paper bag	29.17 ± 6.77 b	55.32	11.11 ± 4.89 b	65.22
	Cloth bag	20.83 ± 6.60 b	68.09	2.08 ± 4.76 b	93.49
2	Untreated	62.50 ± 7.05 a	-	26.39 ± 4.90 a	-
	Paper bag	30.56 ± 6.54 b	51.10	6.94 ± 4.55 b	73.70
	Cloth bag	25.00 ± 6.45 b	60.00	2.08 ± 4.49 b	92.12

^1^ Within a mango orchard, means within a treatment category followed by the same letter are not significantly different, *p* < 0.05, Tukey’s HSD test.

**Table 2 insects-12-00500-t002:** Changes in weight, size, and sphericity of mangoes using different bagging materials for the first and second mango orchards.

Mango Orchard ^1^	Treatment	Weight (g)	Size (cm^3^)	Sphericity
1	Untreated	469.96 ± 9.15 a	94.83 ± 0.59 a	0.469 ± 0.003 a
	Paper bag	454.73 ± 9.03 a	94.51 ± 0.58 a	0.471 ± 0.003 a
	Cloth bag	471.68 ± 8.80 a	95.50 ± 0.56 a	0.471 ± 0.003 a
2	Untreated	280.95 ± 6.22 b	79.74 ± 0.57 b	0.399 ± 0.003 a
	Paper bag	312.23 ± 5.77 a	82.19 ± 0.53 a	0.396 ± 0.003 a
	Cloth bag	307.92 ± 5.69 a	82.28 ± 0.52 a	0.399 ± 0.003 a

^1^ Within a mango orchard, means within a treatment category followed by the same letter are not significantly different, *p* < 0.05, Tukey’s HSD test.

**Table 3 insects-12-00500-t003:** Changes in peel (*L********
*a********
*b******** values) and internal flesh color of mangoes using different bagging materials for the first and second mango orchards.

Mango Orchard ^1^	Treatment	Peel Color			Hue Angle (°)	Chroma	Flesh Color
		*L**	*a**	*b**			
	Untreated	46.77 ± 0.43 b	11.81 ± 1.03 a	10.64 ± 0.54 b	0.772 ± 0.108 a	16.77 ± 0.72 a	2.67 ± 0.12 b
1	Paper bag	49.78 ± 0.43 a	8.81 ± 1.02 a	13.83 ± 0.53 a	0.550 ± 0.107 a	17.99 ± 0.71 a	2.85 ± 0.12 ab
	Cloth bag	48.92 ± 0.42 a	10.93 ± 1.01 a	12.71 ± 0.52 ab	0.646 ± 0.106 a	17.69 ± 0.70 a	3.24 ± 0.11 a
	Untreated	40.59 ± 0.72 b	15.04 ± 1.52 a	9.36 ± 1.34 b	0.395 ± 0.083 a	19.41 ± 1.62 ab	2.74 ± 0.22 a
2	Paper bag	43.26 ± 0.67 a	16.41 ± 1.41 a	14.84 ± 1.24 a	0.627 ± 0.077 a	23.64 ± 1.50 a	3.14 ± 0.20 a
	Cloth bag	41.41 ± 0.66 ab	12.85 ± 1.39 a	10.31 ± 1.22 b	0.528 ± 0.076 a	17.90 ± 1.48 b	2.54 ± 0.20 a

^1^ Within a mango orchard, means within a treatment category followed by the same letter are not significantly different, *p* < 0.05, Tukey’s HSD test.

**Table 4 insects-12-00500-t004:** Economic analysis of pre-harvest bagging against *A. tubercularis* for the first and second organic mango orchards.

Mango Orchard	Treatment	Marketable Yield (kg/ha)	Gross Return (EUR/ha)	Cost of Treatment (EUR/ha)	Net Return (EUR/ha)	Adjusted Net Return (EUR/ha)
	Chemical control	21,000.00 *	37,597.20	525.00	37,072.20	
1	Paper bag	20,319.45	39,952.16	2691.61	37,260.55	188.35
	Cloth bag	21,076.86	43,048.20	3766.28	39,281.93	2209.73
	Chemical control	20,000.00 *	48,159.59	525.00	47,634.59	
2	Paper bag	22,226.73	58,305.27	4288.00	54,017.27	6382.68
	Cloth bag	21,919.91	58,679.25	6000.05	52,679.20	5044.61

* Average production for mango Osteen (21 tonnes/ha) and Sensation (20 tonnes/ha) in Southern Spain [32]. Cost of insecticide (Paraffinic oil 79% (EC) p/v): 3.5 EUR/L (price of product obtained from distributors). Volume of treatment required for a mango orchard: 2000 L/ha. Cost to spray: two laborers/spray = 100 EUR/ha. Cost of paper bag: 0.04 EUR/piece. Cost of cloth bag: 0.07 EUR/piece. Cost of bagging: one laborer/ha = 50 EUR/day. Daily number of bags placed per laborer with experience = 3500 bags/day [26]. Average price received by the farmer for organic mango Osteen cv.: Category I = 2.06 EUR/kg and Category II = 1.22 EUR/kg. Average price received by the farmer for organic mango Sensation cv.: Category I = 2.70 EUR/kg and Category II = 1.59 EUR/kg. To carry out the calculations, Category I is associated to fruit that satisfied the quality parameters established by Spanish marketing companies for export and fresh consumption, and Category II is associated to fruit that exceeded the economic injury threshold (4 pink blemishes per piece) and is destined for juice and other uses. Treatments: Chemical control = foliar spray of paraffinic oil 79% (EC) p/v (Citrol-ina^®^, Sipcam Inagra, SA, USA), dose = 1 L/hL (recommended by the product label) and three times per crop season; Paper bag = bagging by yellow satin paper bag at 71 days before mango harvest; and Cloth bag = Bagging by white muslin cloth bag at 91 days before mango harvest.

## Data Availability

The data presented in this study are available on request from the corresponding author.

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
