# Peer review of "Influence of Pre-Harvest Bagging on the Incidence of Aulacaspis tubercularis Newstead (Hemiptera: Diaspididae) and Fruit Quality in Mango"

_insects, 2021, doi:10.3390/insects12060500_

Round 1
Reviewer 1 Report
The paper reports important information about the use of fruit bagging technique as alternative control method for an important pest of mango, the scale A. tubercularis. Moreover, the authors also report the influence of such technique on other quality parameters of the fruit. This last aspect was in some way studied also by other authors, so is not totally new. Overall the research is interesting and well conducted. English is fair. Some detail is missing in MM section. Discussion should be shortened and improved as some sentences are a little cumbersome.
Line 86-89, unclear, suggest dividing in two sentences
Line 130 I find this part rather weak and needful of more details and specifications.
Line 157-159, what the authors mean with “females colonies “ or “male colonies”? 1 colony = 1,2,3… or more individuals?
Line 309-325. In discussion, I would rather start discussing the results obtained in this study, comparing them with other one obtained from other researchers.
Line 326 “There are no published data on the control of A. tubercularis by using fruit bagging” this was pointed out already in introduction.
Line 329-331, this sentence is cumbersome.
Line 342-348, I find this part a little unclear
Author Response
Line-specific comments:
Line number |
Changes |
83 |
The sentence has been divided in two sentences |
127-135 |
This section describes the morphological and structural characteristics of the two types of bags used in the test. The authors believe that the description is adequate and in accordance with the information provided by the manufacturers of the bags, whose identity has been included in the text. |
156 |
“male colonies and females” changed to “females and male colonies” A. tubercularis male crawlers are settled in colonies of 10–80 individuals, often around the female mother. |
306-331 |
The authors consider appropriate to begin the discussion by mentioning the pests that can cause damage to mango fruit, as well as the varieties most grown in southern Spain. We believe that this information is useful to the reader for a better understanding of the results and discussion presented in this report. |
323-330 |
This part has been rewritten for a better understanding. |
Reviewer 2 Report
This paper evaluates bagging mango fruits for insect protection vs a non-treated control. Fruit bagging is increasingly used to manage pests in fruit production, so this work is part of a broader body of literature. Design and rigor are generally good, I have some question about potential statistical power that can probably be cleared up by better describing the design. The economics section is novel but relatively weak, as the non-bagged, sprayed treatment relies on assumed data that makes for difficult comparison with the empirical results observed for the bagged treatments.
Excellent Introduction. This is very clear and makes the case for both the efficacy of bagging mangoes and the need for this specific work. Specific editorial suggestions, by line:
15: “despite of the bagging proved efficacy” is a clunky phrase. Try “despite the proves efficacy of bagging…”
18: “…types of bags…”
19: “…several other fruits…”
Keywords: fruit bagging; Aulacaspis tubercularis; I wouldn’t include words that are in the title also in the keywords.
49: “being India India being…”
60: …life live plants, have has…”
94: “…reduced up to 100% by covering…”
97: “…90% in sanitation by control of anthracnose…”
113-114: “…mango bagging manage could...”
Materials and methods:
122: I think your between rows and between trees measurements are backwards, should be 4 m between rows and 3 m between trees.
125: Were fungicides and other materials used certified for organic production?
Fruit bags: please list manufacturer
Cluster effect: did this mean that you assigned treatments to groups of trees, but collected data from a central tree to reduce edge effect from other treatments?
Statistical design: it is unclear, did you have n=12 replications for each treatment? And only four observations per replicate? That seems like a lot of replications but pretty low number of observations. The low number of observations could make small differences artificially large when estimating a parameter for the overall population.
Treatments: no chemical control standard?
162: can you cite the specific marketing company standard or point to another published standard?
Fruit quality measurements: Are these standard for mango research? Can you cite a paper that used them?
192-194: This methodology for cost of chemical treatment can lend itself to some pretty subjective results. The discussion should include reference to the relatively low confidence of this comparison, unless empirical data from the same orchards is used for comparison.
Results section is strong.
212: “…significantly provided an increased…”
216: “… blemishes on the epidermis that than the…”
223: “…A. tubercularis with respect to the untreated control…”
3.3 Economic analysis: this looks good, but as mentioned above, some reference to relatively low power of this analysis because of assumptions made should be included. For example, without testing a chemical control treatment in these orchards, we do not know the actual yield or quality of the fruit in a managed, non-bagged program.
316: “…damage until up to 30% of the fruits…”
317-18: “which occupies the 7% of the mango cultivated mango area…”
321: “…most more probable…” “…is so likely more useful…”
332: “…resulting from a late bagging time
338: “... being the given that sphericity was not different through among the treatments…”
341: Why might we see an increase in fruit weight from bagging?
388: may increase fruit size
392: “…which sale sell their…”
405: “…who expect to market a more quality environmentally sustainable production…” This sentence is awkward.
Author Response
General comments:
- In our case, 12 mango trees with presence of white mango scale were selected in each orchard. In each tree, 12 fruits were randomly selected: 4 fruits were bagged with paper bags, 4 fruits with cloth bags and 4 unbagged fruits (controls). The choice of trees was conditioned by the aggregate distribution of the pest in the orchard. In this way, we ensured that the selected fruits were exposed to the pest. Experimental design has been rewritten for a better understanding (Lines 137-141)
- No chemical control was carried out on unbagged fruits.
Line-specific comments:
Line number |
Changes |
15 |
“despite of the bagging proved efficacy” changed to “despite the proves efficacy of bagging” |
18 |
“type of fruit bags” changed to “types of bags” |
19 |
“several fruits” changed to “several other fruits” |
38 |
“fruit bagging” and “Aulacaspis tubercularis”deleted |
38 |
“fruit quality” added |
47 |
“being India” changed to “India being” |
59 |
”life plants, have” changed to “live plants, has” |
93 |
“reduced up to 100% covering” changed to “reduced up to 100% by covering” |
94 |
“in sanitation by” changed to “control of” |
111 |
“manage” deleted |
119 |
“4 m between trees and 3 m between rows” changed to “4 m between rows and 3 m between trees” |
122 |
“some fungicides authorized in organic production (sulphur)” added |
129 |
“(Sanidad Agrícola Econex S.L., Murcia, Spain)” added |
133 |
“(Junejour New Life Store, China)” added |
160 |
“[1,14]” added |
168 |
“according to Chonhenchob et al. [21]” added |
183-196 |
The costs of chemical treatment were estimated according to the usual management of A. tubercularis by farmers in the study area. Because this is a theoretical study, we understand that the results can be interpreted subjectively. |
210 |
“an” deleted |
214 |
“that” changed to “than” |
221 |
“with” added |
314 |
“until” deleted |
315 |
“which occupies the 75% of the mango cultivated area” changed to “which occupies 75% of the cultivated mango area” |
321 |
“most” changed to “more” |
321 |
“so” changed to “likely” |
331 |
“from” added |
337 |
“being the sphericity not different through the treatments” changed to “given that sphericity was not different among the treatments” |
387 |
“increases” changed to “may increase” |
391 |
“sale” changed to “sell” |
404 |
“who expect to market a more quality environmentally sustainable production” changed to “sell a high quality, healthy and sustainable fruit in selected markets” |